# Response of Extreme Precipitation to Dust Aerosols in the Tarim Basin under Climate Warming

Ze Chen<sup>1,2,3,4</sup>, Chenglong Zhou<sup>1,2,3,4\*</sup>, JiaCheng Gao<sup>1,2,3,4</sup>, Congzhen Zhu<sup>1,2,3,4</sup>, Meiqi Song<sup>1,2,3,4</sup>, Yu Wang<sup>1,2,3,4</sup>, Yabin Wei<sup>1,2,3,4</sup>, Lu Meng<sup>1,2,3,4</sup>, Mingjie Ma<sup>1,2,3,4</sup>, and Cong

Wen<sup>1,2,3,4</sup>

<sup>1</sup>Institute of Desert Meteorology, China Meteorological Administration, Urumqi 830002, China

<sup>2</sup>National Observation and Research Station of Desert Meteorology, Taklimakan Desert of Xinjiang, Urumqi 830002, China

3 Taklimakan Desert Meteorology Field Experiment Station of China Meteorological Administration, Urumqi 830002, China

<sup>4</sup>Xinjiang Key Laboratory of Desert Meteorology and Sandstorm, Urumqi 830002, China

Correspondence to: Chenglong Zhou (zclidm@163.com)

#### Abstract.

Climate warming is simultaneously intensifying both dust activity and extreme precipitation (EP) in the arid and semi-arid regions. Meanwhile, dust aerosols further influence EP through complex cloud physical processes. However, significant uncertainties remain regarding the modulating role of dust aerosols within the aerosol-cloud-precipitation interaction system. There is a pressing need to quantitatively resolve this complex process to address disaster prevention challenges in arid regions. Based on long-term ground-based observations, satellite data, reanalysis data, and CMIP6 models, this study leverages a systematic analysis to investigate the impact of dust aerosols on EP in the Tarim Basin. Results reveal that dust-related extreme precipitation (D\_EP) accounts for a relatively high proportion of EP (35.52% for frequency and 34.34% for precipitation amount) in the Tarim Basin. Dust weather provides the necessary lifting

for precipitation., while water vapor acts as a limiting factor. Accordingly, dust aerosols enhance precipitation efficiency by increasing cloud particle radius and promoting cloud water path, ice water path, and cloud top height under conditions of sufficient moisture. Furthermore, the regional average contribution of dust aerosols to EP events is quantified as 6.6% using long-term in situ observations. CMIP6 projections indicate that D\_EP events will persist at relatively high values in the near term. This findings reveal that dust aerosols serve as a key regulator of the water cycle in arid regions, providing a new perspective for understanding the mechanisms driving EP.

**Keywords:** Dust aerosol, Extreme precipitation, Dust-cloud-precipitation interactions, Tarim Basin

# 1 Introduction

Under climate warming, dust events are increasing in frequency and intensity as hazardous phenomena that severely impact the natural environment and human activities (Chen et al., 2023; Liu et al., 2022). Dust aerosols not only directly impair ambient air quality—with PM<sub>2.5</sub> pollution responsible for approximately 4.2 million premature deaths annually (Shaddick et al., 2020), but also substantially influence the earth's energy balance and hydrological cycle through aerosol-radiation and aerosol-cloud interactions (Kok et al., 2023; Song et al., 2024; Zhou et al., 2022, 2023). Climate change is amplifying extreme precipitation (EP) in arid regions, particularly the deserts of Northwest China. A significant wetting trend is observed, marked by increased frequency and intensity of summer EP events and associated floods (Li et al., 2019; Li and Yao, 2023; Yao et al., 2022). These patterns suggest a complex interaction between dust activities and EP under climate change (Zhou et al., 2025), which poses significant challenges to disaster prevention and mitigation efforts.

Current research has advanced our understanding of aerosol-cloud-precipitation interactions, yet a significant knowledge gap remains regarding how dust aerosols influence EP in arid regions under a warming climate. Based on multi-source observations and numerical simulations, existing research has preliminarily revealed that dust

https://doi.org/10.5194/egusphere-2025-5307 Preprint. Discussion started: 7 November 2025 © Author(s) 2025. CC BY 4.0 License.

aerosols influence cloud lifetime and structure through cloud microphysical processes (e.g., ice-nucleating effects) and radiative processes (e.g., altering atmospheric stability), thereby exerting complex influences on precipitation. On the one hand, dust aerosols can enhance precipitation by intensifying ice-phase processes and promoting the growth of solid hydrometeors (Dong et al., 2018; Pan et al., 2024). Additionally, through its radiative effects, the absorption of solar radiation by dust layers induces upper-level warming and surface cooling, which collectively accumulate convective instability and ultimately trigger more intense convective precipitation (Guo et al., 2018; Huang et al., 2014). On the other hand, they may also suppress convective development through radiative effects that induce lower-level cooling, enhance environmental wind shear, and reduce convective available potential energy. Furthermore, an increase in cloud particle number concentration may lead to competitive consumption of available moisture, restricting the effective growth of individual cloud droplets and thereby exerting a suppressive effect on precipitation (Leung and Van Den Heever, 2023; Li et al., 2017). In summary, the net impact of dust aerosols on precipitation is strongly dependent on the ambient thermodynamic and dynamic conditions as well as the cloud type (Zhu et al., 2023). However, these studies are often limited by sparse observational data, uncertainties in model parameterizations, and inadequate representation of physical processes, making it difficult to accurately quantify the net effect of dust under varying meteorological conditions and thus hindering a mechanistic understanding and reliable prediction of EP (Li et al., 2024, 2011).

This study aims to quantitatively assess the response of EP to dust aerosols in the Tarim Basin. Firstly, using ground observations and reanalysis data from spring and summer during 1975-2024, we diagnose the climatic characteristics and environmental conditions of D\_EP events. Then we apply the observational minus reanalysis (OMR) approach to quantitatively reveal the net contribution of dust aerosols to EP at a climate observational scale. Finally, we leverage CMIP6 projections to project the future evolution of D\_EP events. The results would advance the understanding of aerosol-cloud-precipitation interactions and provide scientific support for climate risk assessment and

disaster prevention in arid regions.

#### 2 Data and Methods

#### 2.1 Ground observations

Daily observational data from 1975 to 2024 (March–August) were obtained from 27 national meteorological stations in the Tarim Basin, provided by the National Meteorological Information Center of the China Meteorological Administration. The datasets include variables such as weather phenomena, temperature, relative humidity, and precipitation. Any occurrence of floating dust, blowing sand, or dust storm was recorded as a dust event (Cheng et al., 2023).

## 95 2.2 Satellite observations

The CERES\_SYN1deg cloud dataset (2000–2024, March–August) released by the National Aeronautics and Space Administration (NASA) was employed. Derived from observations by the CERES instrument, this dataset benefits from high pointing accuracy, rapid response, and low systematic error in its servo system (Wielicki et al., 1998), ensuring high reliability for cloud parameter retrieval. The dataset has a spatial resolution of 1° and an hourly temporal resolution. Selected variables from this dataset, including cloud top height, liquid water path, ice water path, and cloud water/ice particle radius, analyzed the influence of dust aerosols on cloud microphysical processes.

#### 2.3 Reanalysis data

The dynamical fields and precipitation data were obtained from the ERA5 and its derivative ERA5-Land datasets, respectively, both provided by the European Centre for Medium-Range Weather Forecasts (ECMWF) (Hersbach et al., 2020). With a horizontal spatial resolution of 0.1°, the hourly reanalysis data were processed into daily averages for each March–August period from 1975 to 2024. The analyzed variables included the u- and v-components of wind, vertical velocity, temperature, and precipitation.

# 2.4 Climate projection data

Projections from five global climate models (UKESM1-0, NorESM2-LM, MPI-ESM1-2-LR, MPI-ESM1-2-HR, AWI-ESM-1-REcoM) from the Coupled Model Intercomparison Project Phase 6 (CMIP6) were analyzed under three Shared Socioeconomic Pathway (SSP) scenarios (SSP126, SSP245, and SSP585) to assess changes in D\_EP over the Tarim Basin (O'Neill et al., 2016; Riahi et al., 2017). It is noteworthy that only five to six models satisfy the aforementioned daily scale output specification. The period 1995–2014 served as the historical baseline, while 2021–2100 was designated as the future projection period.

#### 2.5 Definition of EP events

Extreme precipitation (EP) events were defined using the percentile threshold method, whereby daily precipitation values (≥0.1 mm) at each station were arranged in ascending order, and values exceeding a specific percentile were classified as EP (Liu et al., 2013). When an EP event occurs synchronously with a dust weather event, it is defined as a dust-related extreme precipitation (D\_EP) event, with those occurring in the absence of dust events defined as non-dust extreme precipitation (N\_EP) events. While many studies adopt the 90th percentile or higher thresholds (Hu et al., 2021; Ma, 2021), such an approach yields limited samples in arid regions like our study area, where precipitation is scarce. At the 90th percentile, the average annual frequency of EP events per station is only 1.7 events, with D\_EP events occurring merely 0.4 times, resulting in a sample size insufficient for robust statistical inference. Therefore, the 75th percentile was selected as the threshold to meet research needs. This threshold results in 5,611 total EP events, including 1,588 D\_EP events, accounting for 7% of all precipitation events. This choice maintains the extremity of events while significantly improving sample representativeness.

# 2.6 Quantifying the contribution of dust aerosols on EP

The OMR method was employed to quantitatively estimate the contribution of dust aerosols to EP. Initially, the OMR approach developed to quantify urbanization

effects on climate systems (Cai and Kalnay, 2005; Kalnay and Cai, 2003; Yang et al., 2011), it has been subsequently applied to reveal anthropogenic influences on the lower atmosphere and regional climate (Ding et al., 2013, 2016; Huang et al., 2018; Zhao et al., 2014). More recent applications have successfully uncovered aerosol impacts on daily weather forecast errors and effects of biomass burning aerosols on clouds (Ding et al., 2021; Huang and Ding, 2021). Therefore, this method could identifies physical processes inadequately represented in models by calculating differences between observational and reanalysis data.

The precipitation in the ERA5-Land reanalysis data used in this study originates from ERA5 and indirectly benefits from the assimilation of multi-source observations, yet it does not directly incorporate aerosol processes (Muñoz-Sabater et al., 2021). Furthermore, ERA5-Land reanalysis data also contain the inherent model systematic biases along with factors such as local climate, topography, and precipitation intensity (Gomis-Cebolla et al., 2023; Li et al., 2022; Tan et al., 2023). Therefore, by comparing the discrepancies in D\_EP events between observations and reanalysis data, while removing the associated systematic errors from the reanalysis, we can isolate the contribution of dust aerosols to EP.

#### 2.7 Methodology

The evaluation and analysis of the dust aerosol effects on EP comprises four steps (Fig 1). Step 1. Statistical analysis of the spatiotemporal characteristics of EP. This step involves analyzing the changes in the frequency and amount of EP, followed by identifying the proportion and relationship of EP events influenced by dust aerosols relative to the total EP. Step 2. Uncovering the physical mechanisms through which dust aerosols affect clouds and precipitation. We analyze the background environmental conditions from the perspectives of dynamic and water vapor conditions, and investigate the internal pathways of influence via cloud microphysical processes. Step 3. Quantifying the contribution of dust aerosols to precipitation. Based on long-term observations from national meteorological stations and ERA5-Land reanalysis precipitation data, the OMR method is employed to quantitatively evaluate the contribution of dust aerosols

to EP. Step 4. Projecting future trends in D\_EP. Utilizing CMIP6 models, future changes in the frequency and amount of D\_EP are projected under three different pollutant emission scenarios.

Figure 1. Technical framework of this study.

# 3 Results

# 3.1 Characteristics of EP in the Tarim Basin

Observational data indicate that a majority of stations in the Tarim Basin show significant increasing trends in the frequency (81%) and amount (74%) of EP events (Figs 2a, 2b). Their nine-year moving averages exhibit significant upward trends at rates of 0.43 d·10a<sup>-1</sup> and 3.31 mm·10a<sup>-1</sup>, respectively (Figs 2c, 2d). On average, EP events account for 24% of the total number of precipitation events but contribute up to 75% of the total precipitation amount. Furthermore, these proportions show significant increasing trends at rates of 1.24%·10a<sup>-1</sup> and 1.18%·10a<sup>-1</sup>, respectively (Figs 2e, 2f), highlighting the dominant role of EP in the regional precipitation structure. This phenomenon may be associated with moister atmospheric conditions and enhanced precipitation efficiency under global warming (Zhang et al., 2023).

Figure 2. Spatiotemporal characteristics of EP events in the Tarim Basin during spring and summer, 1975–2024. (a) Spatial distribution of trends in the frequency of EP (units: d·10a<sup>-1</sup>); (b) Spatial distribution of trends in the amount of EP (units: mm·10a<sup>-1</sup>); (c) Interannual variation in the frequency of EP (units: d); (d) Interannual variation in the amount of EP (units: mm); (e) Interannual variation in the proportion of EP frequency to total precipitation frequency (units: %); (f) Interannual variation in the proportion of EP amount to total precipitation amount (units: %). The plus sign (+) and dot (·) in (a) and (b) denote trends that are statistically significant and non-significant at the 95% confidence level based on the Student's t-test, respectively; the asterisk (\*) in (c)–(f) indicates statistical significance.

https://doi.org/10.5194/egusphere-2025-5307 Preprint. Discussion started: 7 November 2025 © Author(s) 2025. CC BY 4.0 License.

To investigate the impact of dust aerosols on EP, a statistical analysis was conducted on EP events associated with dust events in the region. Over the past 50 years, the number of D\_EP days has shown a significant positive correlation with the total number of EP events days (R = 0.64\*; Fig. 3a), while the correlation for precipitation amounts is even stronger (R = 0.65\*; Fig. 3b). Overall, the multi-year average contribution of D\_EP frequency (amount) to total EP is 35.52% (34.34%). Before the year of 2000, D\_EP events were dominant at 21% of the stations, with contribution rates exceeding 90%, a phenomenon particularly pronounced in areas south of TZ Station (Figs. 3c, 3d). The more pronounced contribution of D\_EP events in the southern region may be attributed to two factors: firstly, the spatial pattern of summer moisture transport in the Tarim Basin, which is greater in the southwest and lesser in the northeast (Ma et al., 2025a); secondly, the basin's dust activity exhibits a spatial distribution of higher frequency in the south and lower frequency in the north, providing more abundant aerosol conditions in the southern area (Cheng et al., 2023).

Figure 3. Statistical relationship between D\_EP and total EP during spring and summer, 1975–2024. (a) Scatter plot and fitted line of D\_EP frequency versus total EP frequency (units: d). (b) Scatter plot and fitted line of D\_EP amount versus total EP amount (units: mm). (c) Heatmap of the interannual variation in the percentage contribution of D\_EP frequency at each station (units: %). (d) Heatmap of the interannual variation in the percentage contribution of D\_EP amount at each station (units: %). The asterisk (\*) in (a) and (b) indicates statistical significance.

# 3.2 Impact Mechanisms of Dust Aerosols on EP

To reveal the underlying causes of D\_EP events, the following section will systematically analyze key influencing factors, including dynamic conditions, moisture conditions, and cloud microphysical characteristics.

#### 3.2.1 Dynamic Conditions

Given the vast area of southern Xinjiang, applying a regional average within a geographically fixed coordinate system to spatially dispersed rainfall centers can introduce significant biases. To address this limitation, the study area was reconstructed using a moving coordinate system centered on the stations experiencing D\_EP events, thereby more accurately capturing the dynamic characteristics of the core rainfall region (Xue et al., 2025).

The synergistic effect of dynamic and thermodynamic processes establishes a weather background conducive to precipitation. One day prior to the occurrence of D\_EP events, an ascent center is observed north of the station at the 500 hPa level, while at the 850 hPa level, the ascent center is located south of the station (Figs. 4a, 4b). This vertical misalignment between the upper- and lower-level ascent centers favors the development of a baroclinically unstable atmospheric stratification, thereby providing favorable dynamic lifting conditions for the onset of D\_EP events. Simultaneously, the low-level (850 hPa) horizontal wind field exhibits convergence features near the station, and a local warm core supplies thermodynamic instability energy for convective development, which collectively sustain and intensify the unstable conditions in the region (Fig. 4c).

Figure 4. Dynamic conditions for D\_EP events during spring and summer from 1975 to 2024, composited using a moving coordinate system. (a) Vertical velocity at 500 hPa (units: Pa s<sup>-1</sup>). (b) Vertical velocity at 850 hPa (units: Pa s<sup>-1</sup>). (c) Thermodynamic field at 850 hPa: vectors represent wind field (units: m s<sup>-1</sup>), and shading represents temperature field (units: °C). The star symbol (★) denotes the origin of the composited stations.

# 3.2.3 Moisture Conditions

Humidity is a key limiting factor for the formation of D\_EP events in the Tarim Basin. Although the regional atmospheric total precipitable water vapor (TPW) shows a consistent increasing trend (Fig. 5a), observational data indicate that 78% of the stations experienced a significant decrease in relative humidity (RH; Fig. 5b), while 89% of the stations recorded a significant increase in saturation specific humidity (qs; Fig. 5c). Under climate warming, arid regions in northern China exhibit a pronounced warming trend (Huang et al., 2016). Rising temperatures lead to an increase in saturation vapor pressure, which enhances the air's capacity to hold moisture. Consequently, despite the overall increase in TPW, RH decreases, resulting in a reduction in actual atmospheric moisture levels and making it more difficult for the air to reach saturation and condensation conditions (Figs. 5d, 5e). However, observations show an increasing trend in EP events within the Tarim Basin (Fig 2), which undersores that dust aerosols play a significant role in EP events.

Figure 5. Characteristics of moisture condition variations during spring and summer, 1975–2024. (a) Spatial distribution of the interannual trend in TPW (black dots denote observational station locations; units:  $mm \cdot a^{-1}$ ). (b) Trend in relative humidity (units:  $\% \cdot 10a^{-1}$ ). (c) Trend in saturation specific humidity (units:  $g \cdot kg^{-1} \cdot 10a^{-1}$ ). (d) Scatter plot and linear fit of relative humidity versus temperature. (e) Scatter plot and linear fit of saturation specific humidity versus temperature. The shaded areas in (a), and the plus signs (+) in (b) and (c), indicate statistical significance at the 95% confidence level, whereas the dots (·) denote non-significant trends.

Figure 6 compares the normalized distributions of the difference between RH and

qs across three types of events, providing further insight into the moisture conditions associated with different events. The results show that for D\_EP events, the RH at all stations is significantly higher than during Dust events, with 85% of the stations more prone to saturation. Compared to N\_EP events, although the RH at all stations is relatively lower, 52% of the stations still exhibit a relatively higher tendency toward saturation. The results indicate that compared to Dust events, both D\_EP events and EP events exhibit significantly more favorable moisture conditions. Furthermore, when comparing D\_EP events and EP events, the former demonstrates a higher level of moisture saturation, providing more favorable conditions for dust aerosols to act as ice nuclei or condensation nuclei and thereby promote precipitation.

Figure 6. Normalized differences in RH and q<sub>s</sub> for spring-summer D\_EP events, Ds (Dust events), and EP events during 1975–2024. Black dots represent mean values, red solid lines indicate medians, and gray dashed lines denote the zero reference.

#### 3.2.4 Cloud Microphysical Properties

Satellite observations reveal that D\_EP events exhibit more intense vertical development, as evidenced by higher cloud top heights (CTH) compared to N\_EP events (Fig 7a) This is accompanied by a significant increase in cloud hydrometeor content, with 63% of stations showing a larger liquid water path (LWP) and 85% of stations

showing a larger ice water path (IWP; Figs. 7b, 7c). This enhancement can be attributed to the role of dust aerosols acting as efficient ice nuclei. Their insoluble surfaces provide a solid-liquid interface that facilitates the formation of ice embryos, thereby promoting the freezing of supercooled droplets (Zhang et al., 2012). The resulting ice crystals then grow through processes such as collision-coalescence, significantly increasing the particle size of hydrometeors such as liquid water radius (LWR) and ice water radius (IWR; Figs. 7d, 7e). This ultimately enhances precipitation efficiency by increasing the sedimentation velocity of precipitation particles. These microphysical processes can suppress the formation of light precipitation while promoting the occurrence of heavy precipitation, consequently leading to an increased frequency of extreme weather events (Shao et al., 2022; Zhao et al., 2025).

Figure 7. Differences in cloud microphysical parameters between D\_EP events and N\_EP events during spring and summer, 2000–2024. (a) CTH difference (units: m). (b) LWP difference (units: g m<sup>-2</sup>). (c) IWP difference (units: g m<sup>-2</sup>). (d) LWR difference (units: μm). (e) IWR difference (units: μm). Black dots represent mean values, red solid lines indicate medians, and gray dashed lines denote the zero reference.

In summary, Figure 8 illustrates the mechanisms by which dust influences EP events. Dust events provide strong dynamic lifting conditions. When moisture is insufficient, abundant dust aerosols acting as condensation nuclei "compete" for the limited available water vapor. This increases cloud droplet number concentration while reducing the cloud droplet effective radius, and may shorten cloud lifetime by enhancing droplet evaporation, ultimately suppressing effective precipitation and resulting in a pure dust event (Fig 8a). In contrast, under sufficient moisture conditions, the hygroscopicity of dust aerosols promotes the hygroscopic growth of cloud droplets, and their efficacy as ice nuclei activates ice crystal processes. These ice crystals then grow by deposition and riming, releasing latent heat and accelerating particle collision-coalescence, thereby fostering cloud vertical development and precipitation formation (Figure 8b).

Figure 8. Mechanisms of dust aerosol effects on clouds and precipitation (TB, Tarim Basin; CWP, cloud water particles; IP, ice particles; PRE, precipitation; DA, dust aerosols; AM, ascending motion; CTH, cloud top height; LWP, liquid water path; IWP, ice water path; LWR, liquid water radius; IWR, ice water radius).

#### 3.3 Contribution of Dust Aerosols to EP and Future Projections

Previous studies have confirmed that the OMR method can effectively reveal aerosol effects not captured by reanalysis data (Ding et al., 2021; Huang et al., 2020; Huang and Ding, 2021). A broad consensus supported by robust theoretical evidence indicates that aerosol-radiation interactions and aerosol-cloud interactions are key drivers of precipitation changes (Zhao et al., 2024). Whether these interactions result in an enhancement or suppression of precipitation depends on the specific physical mechanisms dominated by dust aerosols. Therefore, as shown in Fig 9a, the probability density distribution of the dust aerosol contribution to EP, derived from bias-corrected ERA5-Land precipitation data, spans both negative and positive values, but exhibits a unimodal pattern with a peak interval of 5% to 25%. From a net effect perspective, the mean contribution of 6.6% demonstrates an overall positive influence of dust aerosols on regional EP.

To further understand the impact of dust aerosols on EP, this study analyzes CMIP6 model projections of future changes in D\_EP events under different emission scenarios. The annual frequency of D\_EP days in the Tarim Basin region shows distinct differences across scenarios: it exhibits a significant decreasing trend under the SSP126 scenario, while the trend is weakest and statistically non-significant under the SSP245 and SSP585 scenario, maintaining high-level fluctuations in the near-term (2021–2040) (Fig 9b). The projected trends for D\_EP events amount are similar to those for frequency, showing significant decreasing trends across all three scenarios, with a more gradual decline under the high-pollution scenario (Fig 9c).

Figure 9. Contribution of dust to EP and projected future changes in D\_EP events. (a) Probability density distribution of the dust contribution rate to EP. (b) Time series of the annual D\_EP days from historical observations and multi-scenario future projections (units: d). (c) Time series of the total annual D\_EP events from historical data and multi-scenario future projections (units: mm).

# 4 Discussion and Conclusion

This study indicates that the actual atmospheric humidity in the study region has

https://doi.org/10.5194/egusphere-2025-5307 Preprint. Discussion started: 7 November 2025 © Author(s) 2025. CC BY 4.0 License.

decreased despite increased water vapor content under climate warming, making saturation more difficult to achieve. This conclusion aligns with the analysis of moisture conditions during the Meiyu period by Sun et al. (2023). However, the occurrence of dust events not only provides favorable dynamic conditions for precipitation but also supplies abundant aerosols (Ma et al., 2023, 2025b; Meng et al., 2025; Zhou et al., 2020, 2025). Dust aerosols can act as both cloud condensation nuclei and ice nuclei, thereby influencing clouds and precipitation. On one hand, they can increase cloud particle number concentration, competing for available water vapor and potentially reducing precipitation. On the other hand, the dusty environment also favors the rapid growth of precipitation particles and can enhance precipitation by promoting water vapor saturation (Li et al., 2024; Sun et al., 2022; Zhu et al., 2023). Observational evidence confirms that dust aerosols are a significant factor affecting the regional water cycle. In specific years at certain stations, their contribution to EP events exceeds 90%, highlighting their crucial role in the formation of EP events.

CERES satellite observations reveal that dust aerosols lead to increases in both the number concentration and size of cloud droplets and ice crystals. In practical observations, the physical characteristics of clouds or precipitation are influenced by both aerosol indirect effects and atmospheric thermodynamic and dynamic effects. Isolating these two types of effects represents the most challenging aspect in observational studies of cloud-aerosol interactions (Zhu et al., 2023). The OMR method can quantify physical processes not fully represented in models while, by focusing the analysis on specific precipitation events, effectively mitigating potential biases caused by systematic differences between observations and reanalysis data on climatic scales (Wang et al., 2013). Results show that the average net contribution of dust aerosols to EP events is 6.6%. Negative contributions at some stations may also originate from the accumulation of absorbing aerosols in the lower layers, which can intensify near-surface evaporation through warming and drying. This evaporation process, governed jointly by temperature, humidity, and aerosol content, exerts a greater influence on near-surface precipitation rates than other factors (Sun et al., 2023). Nevertheless, since dust aerosols

https://doi.org/10.5194/egusphere-2025-5307 Preprint. Discussion started: 7 November 2025 © Author(s) 2025. CC BY 4.0 License.

generally exert a promotive effect on precipitation, D\_EP events will persists at a high level in the near-term 21st century under high-emission scenarios. This study demonstrates that ecological engineering for windbreaking and sand fixation also delivers significant benefits for flood prevention and disaster mitigation.

Code availability. The data and data analysis method are available upon request.

Data availability. Meteorological observation data were provided by the National Meteorological Information Center at http://data.cma.cn under a restricted license and are thus not publicly distributable. The CERES dataset can be obtained from https://ceres.larc.nasa.gov/Data/. The ERA5 reanalysis dataset is accessible via the ECMWF at https://cds.climate.copernicus.eu/datasets/reanalysis-era5-pressure-levels?tab=overview. Daily precipitation data from five CMIP6 models are available through the Earth System Grid Federation's CMIP6 archives at https://esgf-node.llnl.gov/projects/cmip6/.

Author contributions. Ze Chen conducted the analysis and validation, created the visualizations, and wrote the original draft of the manuscript. Chenglong Zhou designed the study and contributed key ideas as well as the review and revision of the manuscript. JiaCheng Gao and Congzhen Zhu provided the data. Meiqi Song, Yu Wang and Yabin Wei interpreted the data. Lu Meng, Mingjie Ma, and Cong Wen contributed to the interpretation and writing of the paper with contributions from the coauthors.

**Competing interests.** The contact author has declared that none of the authors has any competing interests.

**Acknowledgements.** The authors are grateful to the science teams for providing the accessible data products used in this study.

20

420

430

435

Financial support. This work was jointly supported by the Tianshan Young Talent Support Program of Xinjiang Uygur Autonomous Region (2024TSYCCX0044), Major Science and Technology Program of Xinjiang Uygur Autonomous Region (2024A03006-1), Natural Science Foundation of Xinjiang Uygur Autonomous Region (2022D01A366), Youth Innovation Team of China Meteorological Administration (CMA2024QN13), Scientific and Technological Innovation Team (Tianshan Innovation Team) project (2022TSYCTD0007), the S&T Development Fund of CAMS (2021KJ034), and Tianchi Talent Program of Xinjiang Uygur Autonomous Region (2025).

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
