# Peer review of "Response of Extreme Precipitation to Dust Aerosols in the Tarim Basin under Climate Warming"

_EGUsphere, 2025_

## Referee Comment (RC1)

The interaction of aerosols and precipitation is an important issue in climate change and atmospheric sciences, especially for the arid and semi-arid lands. This article targeted the Tarim Basin with dry to explore the impact of dust aerosols on extreme precipitation over arid region with multi-source observation over past 50 years. The study provides the interesting results on response of extreme precipitation to dust aerosols over desert and surrounding region, improving our understanding on the interaction of aerosols and precipitation over arid region. The conclusions of this article on dust aerosols altering cloud physical properties are innovative to a certain extent. I recommend the article be published after the following comments are addressed.

- The main concerns are around the in-depth discussions and clarification on the results:
  - a) In Section 3.2.1, Dynamic Conditions: the synergistic effect of dynamic and thermodynamic processes establishes strong lifting conditions conducive to both dust event and extreme precipitation, which could ensconce a positive correlation between dust event and extreme precipitation in the macro process. Please give the discussions on this the macro process for response of extreme precipitation to dust aerosols.
  - b) Lines 201-202: Overall, the multi-year average contribution of D\_EP frequency (amount) to total EP is 35.52% (34.34%). Please clarify how to estimate the contribution.
  - c) In Section 3.2.4 Cloud Microphysical Properties: Please add the supplementary with the Figures presenting the absolute values of Cloud physical (not only Microphysical) properties CTH, LWP, IWP, LWR and IWR for the complete discussion on D\_EP events, which could support the mechanisms of dust aerosol effects on clouds and precipitation in Fig. 8.
  - d) It is suggested for the further study for another paper with the CMIP6 model projections of future changes in D\_EP events under future climate. The current discussion and results are too simple and surplus with rough analysis.
  - e) Section 4 Discussion and Conclusion: the discussion is too many with

repeating the discussions in Sect. 3. Please modify this section with 1) concluding the study results, 2) extracting the highlights, 3) discussion on the limitations of this study, and 4) outlook of further study on this issue.

**2. Technical comments:**

- a) It is suggested with the better title: Response of Extreme Precipitation to Dust Aerosols in the Tarim Basin under Climate Warming and Wetting or Response of Extreme Precipitation to Dust Aerosols in the Tarim Basin over past 50 years
- b) Please label the station names with station number in y-coordinate the Figure 3c-d
- c) Please give the full names of acronyms at the first appearance of acronyms, such as OMR, SSP245 and SSP585.
- d) Please change "3.2.4 Cloud Microphysical Properties" to "3.2.4 Cloud physical Properties", because of some 3cloud macrophysical Properties.
- e) Please unify "South Xinjiang" with "Tarim Basin" in the manuscript.